# Association between Expression of Connective Tissue Genes and Prostate Cancer Growth and Progression

**DOI:** 10.3390/ijms24087520

**Published:** 2023-04-19

**Authors:** Patrick-Julien Treacy, Alberto Martini, Ugo Giovanni Falagario, Parita Ratnani, Ethan Wajswol, Alp Tuna Beksac, Peter Wiklund, Sujit Nair, Natasha Kyprianou, Matthieu Durand, Ashutosh K. Tewari

**Affiliations:** 1Department of Urology, Icahn School of Medicine at Mount Sinai, New York, NY 10029, USA; 2Department of Urology and Organ Transplantation, Nice University Hospital, 06003 Nice, France; 3Department of Urology, Vita-Salute San Raffaele University, 20132 Milan, Italy; 4Department of Urology and Organ Transplantation, University of Foggia, 71122 Foggia, Italy; 5Department of Oncological Sciences, Icahn School of Medicine at Mount Sinai, New York, NY 10029, USA

**Keywords:** prostate, prostate cancer, connective tissue, genetic, transcriptomic, Decipher, outcomes, TCGA, collagen, fibronectin, extra-capsular extension

## Abstract

To find an association between genomic features of connective tissue and pejorative clinical outcomes on radical prostatectomy specimens. We performed a retrospective analysis of patients who underwent radical prostatectomy and underwent a Decipher transcriptomic test for localized prostate cancer in our institution (*n* = 695). The expression results of selected connective tissue genes were analyzed after multiple *t* tests, revealing significant differences in the transcriptomic expression (over- or under-expression). We investigated the association between transcript results and clinical features such as extra-capsular extension (ECE), clinically significant cancer, lymph node (LN) invasion and early biochemical recurrence (eBCR), defined as earlier than 3 years after surgery). The Cancer Genome Atlas (TCGA) was used to evaluate the prognostic role of genes on progression-free survival (PFS) and overall survival (OS). Out of 528 patients, we found that 189 had ECE and 27 had LN invasion. The Decipher score was higher in patients with ECE, LN invasion, and eBCR. Our gene selection microarray analysis showed an overexpression in both ECE and LN invasion, and in clinically significant cancer for *COL1A1*, *COL1A2*, *COL3A1*, *LUM*, *VCAN*, *FN1*, *AEBP1*, *ASPN*, *TIMP1*, *TIMP3*, *BGN*, and underexpression in *FMOD* and *FLNA*. In the TCGA population, overexpression of these genes was correlated with worse PFS. Significant co-occurrence of these genes was observed. When presenting overexpression of our gene selection, the 5-year PFS rate was 53% vs. 68% (*p* = 0.0315). Transcriptomic overexpression of connective tissue genes correlated to worse clinical features, such as ECE, clinically significant cancer and BCR, identifying the potential prognostic value of the gene signature of the connective tissue in prostate cancer. TCGAp cohort analysis showed a worse PFS in case of overexpression of the connective tissue genes.

## 1. Introduction

In the United States, prostate cancer is the second most prevalent cause of cancer death in men, and most diagnosed, with an incidence of 230,000 cases yearly [1], and more than 25,000 deaths per year. Six out of ten men older than 65 years old will eventually be diagnosed, with the average age being about 66 at time of diagnosis [2]. Prostate cancer is determined using the Gleason grading system, which gives a prognostic value to both pathologic and clinical outcomes. This heterogeneous disease can lead to various postoperative outcomes, with differences in clinical features such as extra-prostatic extension, lymph node invasion and early biochemical recurrence in patients with the same Gleason score, significantly reducing both overall survival and progression-free-survival (PFS). Prostate cancer patients receive inappropriate treatment in over 50% of cases due to a misclassification and suboptimal risk stratification [3], resulting in overtreatment, delayed adjuvant therapy and leading to poor outcomes, financial drain and exposure to adverse treatment effects [4]. To date, various drivers of aggressiveness in prostate cancer have been described, not only glandular but also on the stromal, tumor micro environment (TME) [5] and genomic aspect [6], refining prognosis or risk of early biochemical recurrence after radical treatment option. Due to the heterogeneous nature of prostate cancer, inadequate tissue sampling during prostate biopsy, imperfect visualization of cancer with MP MRI and histological grading system limited to glandular architecture (ignoring stromal features), risk stratification could be improved by adding additional features such as TME. Treatment options can range from active surveillance to focal therapy, surgery and radiotherapy for localized disease, while biochemical recurrence and metastatic cancers are treated using either hormonotherapy or chemotherapy.

The prostate gland primarily consists of gland and stroma. The normal prostate gland has papillary projections and is composed of two layers of cells: inner columnar epithelia and outer cuboidal basal cells. The stromal part is a collection of loose connective tissue consisting of collagen fibers (main structural and mechanical component) along with fibroblasts, auto fluorescent myoblasts, elastin fibers, vasculature, nerves, and immune components [7]. During prostate cancer development, tumor cells grow in the prostate glands, surrounded by TME consisting of myofibroblasts, fibroblasts and endothelial cells defined as the reactive stroma, and both adaptive and innate immune cells. Cytokines and extracellular molecules are secreted by the reactive stroma in a dynamic cross-talk with the tumor environment, and together create the extracellular matrix (ECM), regulating both proliferation and migration of tumor cells [8], with this leading towards phenotypic progression of tumor invasion promoting angiogenesis and tumorigenesis [9]. Breaching of cancer cells through basal membrane leads to an automatic stromal reaction. Connective fibers, such as collagen, vimentin, laminin, and fibronectin, have already been shown to contribute to the development of various cancers, including breast cancer [10,11], bladder cancer [12], and pancreatic cancer [13], as well as prostate cancer through the matrix and cell-to-cell adhesion, helping tumor cells to escape dormancy. Collagen is one of the most important components of ECM, and fibronectin has a critical role in cell adhesion, being able to attach tumor cells to collagen and other fibers of the ECM [14], fighting against anoikis [15]. The activation of MAPKs regulating tumor development is indeed due to adhesion to the extracellular matrix [16], contributing to a cascade of events, ultimately resulting in tumor cells leaving the local site and leading to metastases in distant sites of the body, developing resistance to anoikis [17]. These features can also be seen on Brightfield microscopy using previously published immunohistochemical markers [5].

Several genetic tests based on cutting-edge technology have been used in order to predict derogatory outcomes, either in DNA sequencing (Sema4) [18] or in RNA expression (Decipher, Prolaris) [19,20,21].

This study evaluated the association of several connective tissue gene expressions known to have a prognostic signature in other cancers and poor postoperative clinical outcomes, such as clinically significant cancer, lymph node invasion, extra-prostatic extension and early biochemical recurrence found in our cohort. The association between an overexpression of our selected genes and worst prognosis regarding overall survival (OS) and progression-free-survival (PFS) defined the potential prognostic value of the connective-tissue-dominated phenotype with clinical tumor progression.

## 2. Results

### 2.1. Clinicopathological Characteristics of Patients

A total of 528 patients, with at least a 3-year follow-up, that underwent Decipher score were included in the study. The median PSA value was 6 ng/mL and median age before surgery was 64 years old. Overall, 27 patients had an ECE (5.1%), 19 had a LN invasion (3.6%), and 69 patients had an early BCR (13.1%).

### 2.2. Decipher Risk Category and Postoperative Features (ECE, LN Invasion, ISUP and eBCR)

Table 1 shows the odds ratio of going into intermediate or high Decipher risk category, which was significantly higher in ISUP 4 and 5 populations, as well as in early BCR.

### 2.3. Gene Expression Correlation with Pejorative Outcomes

#### 2.3.1. ISUP Score

The results shown in Figure 1 first provide a comparison between the ISUP 2 and ISUP 3 populations (panel A), and then associate the ISUP 1 and 2 populations with the ISUP 3 to 5 populations (panel B).

Upon comparative analysis of ISUP 2 vs. ISUP 3, 619 genes were found to be significantly upregulated and 2175 genes downregulated. *COL1A1*, *COL1A2*, *COL3A1*, *VCAN*, *ASPN*, *BGN*, *MME* were significantly upregulated, while FLNA and FMOD were downregulated.

When comparing ISUP 1 and 2 vs. ISUP 3 to 5 populations, 887 genes were upregulated, while 3065 genes were downregulated. *COL1A1*, *COL1A2*, *COL3A1*, *VCAN*, *AEBP1*, *ASPN*, *BGN*, *COMP* and *MME* were significantly upregulated, while *FLNA* and *FMOD* were downregulated.

#### 2.3.2. Lymph Node Invasion

Comparative analysis of the population with or without lymph node invasion revealed an upregulation of 407 genes and downregulation of 707 genes. *COL1A1*, *COL1A2*, *COL3A1*, *LAM* genes, *LUM*, *VCAN*, *FN1*, *AEBP1*, *MAP1B*, *ASPN*, *BGN* and *COMP* were significantly upregulated.

#### 2.3.3. Extra-Capsular-Extension

The data shown on Figure 1D indicate that 501 genes were significantly upregulated in the ECE population compared to the population without ECE, and 932 genes were downregulated. The following genes *COL1A1*, *COL1A2*, *COL3A1*, *LUM*, *VCAN*, *FN1*, *AEBP1*, *MAP1B*, *ASPN* and *COMP* were upregulated, while *FLNA* and *FMOD* were downregulated.

#### 2.3.4. Pathway Analysis of Our Connective Tissue Genes Selection

Using the DAVID bioinformatics software, we found six pathways to be associated with the genes that we previously selected (Table 1), with an enrichment score of 5.54. ECM-receptor interaction, focal adhesion, protein digestion and absorption, amoebiasis, the PI3K-AKT signaling pathway and Platelet activation were all enriched (*p* < 0.001). Collagen, fibronectin, laminin, tenascin and THBS were activated in the ECM-receptor interaction pathway (Appendix A) and the ECM-receptor interaction pathway was active in the focal adhesion pathway, with an associated activation of filamin (Appendix A).

### 2.4. Outcomes Correlation

Table 2 reveals the repartition of genes’ upregulation in the TCGAp population. Of the 491 patients from the TCGAp cohort, 111 patients (22%) had an expression modification in the queried genes. All three collagen genes were upregulated in 3 or 4% of the patients (*p* < 0.05). There was also a significant co-occurrence in most of our selected connective tissue genes (Table 3 and Table 4).

Regarding the progression-free survival (PFS), out of the 111 patients with an expression modification in queried genes, 27 had a disease progression, with an 81.24 median of disease-free months. The 5-year PFS on the population with overexpression of the connective tissue genes was 53.27% vs. 68.2% for the population without overexpression of these genes (log-rank showed a statistical significance (*p* = 0.0315) in the Kaplan–Meier survival curve) (Figure 2).

Shown in Figure 2, progression-free-survival curves of the TCGAp cohort according to population with an overexpression of the connective tissue genes (red population) and population without overexpression of these genes).

## 3. Discussion

The present study demonstrates a significant association between an upregulation in the transcriptomic of connective tissue genes and worst clinical outcomes, such as a higher pathology grade, higher final stage (extra-capsular extension and lymph node invasion). We then validated our results to show a significant association between connective tissue genes dysregulation and progression-free survival. Connective tissue has already been linked to pejorative outcomes in various studies in the past [22]. For example, Ayala et al. showed that prostatic carcinoma generated three patterns of stromogenic carcinoma, correlated with a higher risk of recurrence, regardless of the epithelial grade component [5,23]. Reactive stroma in prostate cancer leads to an upregulation of more than 500 genes, and alterations in novel processes such as neurogenesis, axonogenesis and the DNA repair/damage pathways [24].

Gleason score (GS) focuses on the cancer cell and the global glandular differentiation, ranging from distinctly infiltrative margins and alternating glands in size and shape (GGG 3) to no glandular differentiation and individual cells (GGG 5) in prostate cancer. Various modifications have been performed to GS through the years to refine the diagnosis for a better prognosis. However, these modifications never included the stroma, and studies have shown a prognostic variability between the same GS population [23]. One of the main raisons for that absence is the use of Hematoxylin and Eosin in gold standard pathology diagnosis. While Hematoxylin stains nuclei blue, eosin stains the extracellular matrix and the cytoplasm pink, and other structures take different shades due to the combinations of these colors leading to a difficult distinction between structures such as smooth muscle or myofibroblasts [25].

Improvements of the Gleason grading system continue to have limitations [26]. The complexity of the new Gleason grading system makes it harder for clinicians. Some older Gleason 6, being reclassified as 7, now have a different clinical management, and studies have shown that virtually no pure Gleason score 6 tumors are associated with disease recurrence after radical prostatectomy, and pure Gleason 6 cancer at radical prostatectomy lacks the potential for lymph node metastases [27] Another problem in the modern Gleason grading system is the lumping of Gleason score 7. Whereas many clinicians consider Gleason score 7 on biopsy to be intermediate risk, multiple studies have shown that a Gleason score 4  +  3  =  7 demonstrates worse pathological stage and biochemical recurrence rates than 3  +  4  =  7 [28].

Asporin, one of the connective tissue proteins, has already shown its major role in cancer progression, restricting mesenchymal stromal cell differentiation, altering the tumor microenvironment, and driving therefore a metastatic progression. Vimentin is a structural protein, creating type 3 intermediate filaments that are expressed in mesenchymal cells, as part of their cytoskeleton, and is often used as a marker of cells undergoing an EMT (epithelial–mesenchymal transition) [29]. There is strong evidence to support a major role for vimentin in the regulation of cancer growth, showing a higher protein expression level in metastatic prostate cancer patients with a Docetaxel resistance [30], in the context of the EMT phenotype [31].

Collagen genes were also one of the major upregulated genes in our cohort, especially COL1 and COL3 genes, coding, respectively, for collagen type I and collagen type III, in the vast majority of the outcomes (higher ISUP score, ECE, LNI). Collagen plays an important part in various cancers [32], such as collagen density increasing tumor formation in breast cancers [10], collagen stiffness in the promotion of muscle-invasive bladder cancer [12], and the ratio of isotropic-orientated collagen fibers in the higher ISUP population in prostate cancer [33].

The correlation identified between a significant upregulation in the connective tissue genes and the extra prostatic extension provides support for the concept that the actual physical growth of the cancer lesion, leading to an extra-prostatic extension, would be both the result of the multiplication of cancer cells and of the growth of the connective tissue in the TME, enabling the dissemination of cancer cells. Our findings are in accordance with the proven evidence that ECE has been associated to a bigger risk of BCR [34]. Ruppender et al. also showed that the tumor cells needed a cell–cell contact to proliferate, with an activation of beta1 integrin [35] in prostate cancer.

Our study also identified the enrichment of both ECM-receptor interaction and focal adhesion pathways. Interestingly, previous studies showed that integrin-mediated cellular adhesion to the extracellular matrix triggers MAPKs, regulating the tumor growth [36]. These results gain support from Chen’s work, which previously established an enrichment on ECM-receptor interaction pathway in PCa patients [37].

Moreover, in the TCGAp database, our chosen genes showed to be upregulated in 0.6 to 7% of the patients, with 26% of the population showing an alteration of the gene queried. Patients with an alteration of the connective tissue genes’ expression had a worse PFS than patients without the gene’s alteration in a prostate cancer cohort. These findings are in accordance with Jia’s work, predicting subsequent relapse by the expression changes in the prostate cancer stroma [38].

Our results provide new significant insights in the therapeutic management of prostate cancer patients exhibiting an upregulated connective tissue gene profile. The connective tissue growth factor (CTGF), activating the building of the connective tissue, induces the collagen expression in human stromal cells [39], having an effect on bone metastasis [40]. FG-3019, a human monoclonal antibody against CTGF, is actually under clinical investigation as a therapeutic agent for patients with serous ovarian cancer [41]. However, in our study, we did not find CTGF overexpressed in our cohort. This can be due to the degree of variability of the region of interest selected by the pathologist sent for genomic testing. However, this represents one of the most common limitations in all studies involving tissue for genetic testing.

Additional limitations of our study include the small patient cohort used for the analyses, even though we have RNA expression data for all our patients. To partly address this issue, we validated our selection of connective tissue genes in another cohort to confirm our results in terms of significant endpoints such as the progression free survival, validating the clinical benefit of showing this upregulation in the connective tissue genes on patient with worst outcomes. Another limitation is the “cherry picking” selection on our connective tissue genes, after a literature review which allowed us to highlight connective tissue genes that have been previously involved in solid tumor pejorative outcomes. Finally, we did not obtain experimental data to support and confirm our results, such as through immunohistochemistry or other genomic tests, as the main direction of the publication was a translational approach rather than a pure fundamental approach. One of the perspectives of the study would be to correlate these results with immunohistochemistry or Red Sirius staining on collagen 1 and 3 to confirm our findings on additional experiments regarding lymph node invasion or extra-capsular extension. Another perspective would be to examine whether the overexpression of those genes promotes the invasion and migration ability of prostatic cancer cells.

These findings are of potential clinical significance in the stratification of prostate cancer patients with the distinct connective tissue gene signature during prostate cancer progression. Thus, in the stratification between active surveillance and radical treatment during the localized stage, a closer follow-up after prostatectomy for patients with the high connective tissue gene signature, towards precision medicine with anti-CTGF treatment for locally advanced disease and upregulation of the connective tissue genes, is recommended.

## 4. Materials and Methods

### 4.1. Patients

The study was approved by the Institutional Review Board (n.17-02467), and all patients provided written informed consent. We started implementing the Decipher test as a standard of care on our patients in 2014. We therefore included all men that underwent a radical prostatectomy for localized prostate cancer in our institution between 2014 and 2018, operated on by a single surgeon, and that had received the Decipher test on the final pathology specimen following surgery. We excluded patients that did not have a more than 3-year follow-up in order to determine early biochemical recurrence (eBCR). One uro-pathologist (KGH) reviewed all the pathology slides using the latest International Society of Urological Pathology (ISUP) Consensus [14]) to score the lesions.

### 4.2. RNA Expression Analysis

The Decipher test (GenomeDx Biosciences Laboratory, San Diego, CA, California) was obtained from the highest Gleason grade group lesion of 0.5 cm^2^ formalin-fixed and paraffin-embedded blocks from the radical prostatectomy specimens. The profiling of the gene expression was executed as previously described [6]. The single-channel normalization algorithm was used for probe-set summarizing and normalizing the microarray results [42]. The Decipher test delivers a categorical and continuous variable, reporting a continuous variable score ranging from 0 to 1, indicating a higher probability of metastasis as the score increases. The categorical variable stratifies in three risk categories, using two cut-offs: low risk ranging from 0 to 0.45, intermediate risk ranging from 0.45 to 0.60, and high risk ranging from 0.60 to 1.0 [14]. Raw data from RNA expression in genes selected for Decipher score were extracted for analysis in our study.

### 4.3. Gene Selection

We ran multiple *t* tests regarding different clinical features such as extra-capsular extension (ECE), lymph node invasion (LNI) and higher Gleason group grade (GGG) to compare gene expression between populations with or without these worst clinical features, as a supervised expression analysis on the whole exome. Once we found the genes that were significantly over or under-expressed for each clinical feature, we then selected all the genes contributing to ECM’s connective tissue and previously implicated in tumorigenesis (Table 1). No control was performed regarding gene expression since we collected raw data from Decipher scores regarding gene expression in our patients’ cohort. Pathway association analysis was subsequently performed, using Database for Annotation, Visualization and Integrated Discovery (DAVID) bioinformatics resources 6.8, on our selection of connective tissue genes. (https://david.ncifcrf.gov/home.jsp (accessed on 13 March 2023))

### 4.4. Correlation with Clinical Outcomes

cBioPortal (Ref. [16]) is a Web resource for cancer genomics, showing disease progression data correlated with multidimensional genomic datasets. We used the TCGA provisional (TCGAp) to obtain mRNA expression and survival information. The TCGA provisional (TCGAp) database was used to obtain mRNA expression of selected connective genes and the correlation with both overall survival (OS) and progression-free survival (PFS) of prostate cancer patients.

### 4.5. Statistical Analysis

We initially examined the correlation between the Decipher score (categorical and continuous variable) and worse clinical tumor features (extra-capsular extension (ECE), lymph node invasion (LN+), higher ISUP, biochemical recurrence) by measuring the odds ratio of going into intermediate and high Decipher risk category according to the clinical outcomes using a logistic regression.

Multiple *t* tests were used to assess the significant statistical differences in the whole exome gene’s expression between groups with and without ECE, LN invasion, clinically significant cancer and eBCR. We analyzed the transcriptomic expression of our selected genes between tumors with and without ECE, LN invasion, and clinically significant cancer (ISUP 1 and 2 vs. ISUP 3, 4, 5). The Benjamini–Hochberg method was used for multiple testing adjustment with a false discovery rate of 5%. The Kaplan–Meier method was used for the PFS and the equality of survivor functions used the log-rank test to confirm severe prognosis.

Statistical analyses were conducted using Stata 14 (StataCorp LP, College Station, TX, USA).

The statistical significance among the group was set at *p* < 0.05.

## 5. Conclusions

Our study demonstrates that connective tissue contributes to prostate cancer growth and progression, and overexpression of connective-tissue-associated gene signature correlates with poor outcomes in prostate cancer. Our findings were confirmed in an external cohort in which the overexpression of the connective tissue genes conferred worse PFS.

## Figures and Tables

**Figure 1 ijms-24-07520-f001:**
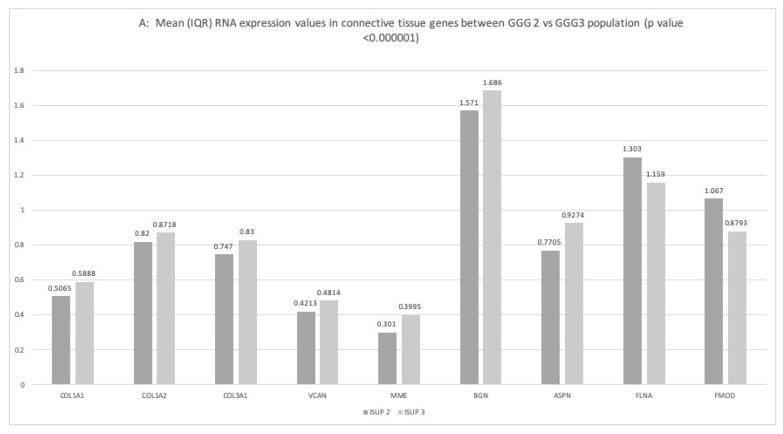
Transcriptomic differences in the connective tissue genes correlated with pejorative outcomes: ISUP 2 vs. ISUP 3 (**A**); ISUP 1 and 2 vs. ISUP 3 to 5 (**B**), Lymph node (LN) invasion or no lymph node invasion (**C**); Extra-capsular extension (ECE) or no (**D**). Statistically significant difference among the different values was set at *p* < 0.05. Mean RNA expression are in measured in IQR.

**Figure 2 ijms-24-07520-f002:**
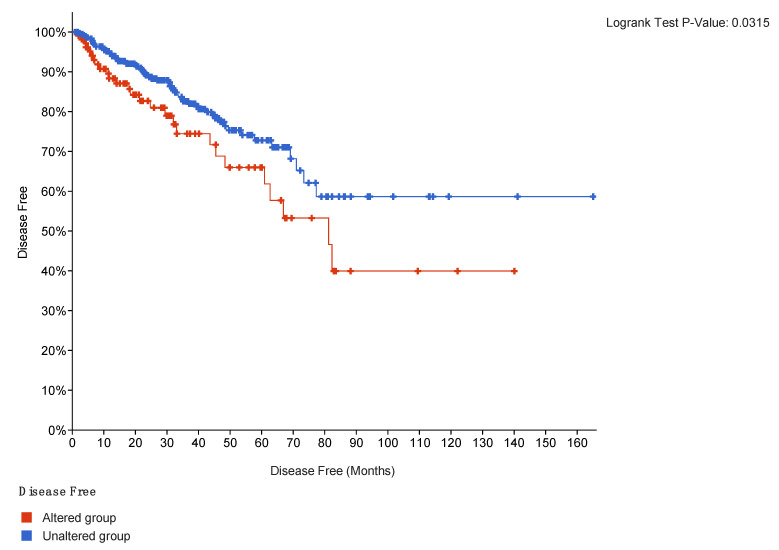
PFS survival was analyzed using Kaplan–Meier curve analysis and compared to patient population harboring alterations in the connective tissue genes (red) vs. the population without any genetic alterations in the connective tissue genes (blue).

**Table 1 ijms-24-07520-t001:** Binary logistic regression predicting Intermediate/High Decipher risk category.

Covariates	Odds Ratio	95% Minimal Confidence Limit	95% Maximal Confidence Limit	*p* Value
ECE	4.154	0.475	36.3	0.1978
LN invasion	0.865	0.061	12.256	0.9149
ISUP 3	2.138	1.392	3.283	0.229
ISUP 4 and 5	9.535	3.649	24.917	0.0001
early BCR	2.115	1.108	4.038	0.0231

**Table 2 ijms-24-07520-t002:** Genetic alteration of our selection of connective tissue genes in the TCGA cohort.

Queried Genes	*COL1A1*	*COL1A2*	*COL3A1*	*LUM*	*VCAN*	*FN1*	*AEBP1*	*MAP1B*	*ASPN*	*COMP*	*MME*
Genetic alteration repartition	3%	4%	4%	5%	3%	5%	5%	5%	4%	4%	4%

**Table 3 ijms-24-07520-t003:** Significant co-occurrence in significant expression modification of the connective tissue genes in patients with our queried genes’ expression modification.

Genes	Co-Occurrence with	Number of Co-Occurrences	*p* Value
*COL1A1*	*COL1A2*	32	<0.01
	*COL1A3*	32	
	*AEBP1*	21	
	*ASPN*	20	
	*COMP*	13	
	*LUM*	11	
*COL1A2*	*COL3A1*	32	<0.01
	*AEBP1*	22	
	*ASPN*	21	
	*VCAN*	15	
	*LUM*	13	
	*COMP*	13	
*COL3A1*	*ASPN*	20	<0.01
	*AEBP1*	19	
	*VCAN*	15	
	*COMP*	13	
	*LUM*	10	
	*MAP1B*	6	
*LUM*	*AEBP1*	8	0.019
	*VCAN*	7	
	*ASPN*	7	
*VCAN*	*COMP*	11	<0.01
	*AEBP1*	11	
	*ASPN*	10	

**Table 4 ijms-24-07520-t004:** Connective tissue genes and function.

Gene Name	Protein	Function
*COL1A1, COL1A2*	Collagen type 1	Most abundant collagen, interstitial matrix component.
*COL3A1*	Collagen type 3	Fibrillar collagen group, mostly found in vascular and microvascular system.
*VIM*	Vimentin	Intermediate filament expressed in mesenchymal cells.
*VCAN*	Versican	Proteoglycan involved in cell adhesion.
*BGN*	Biglycan	Proteoglycan involved in collagen fibril assembly.
*ASPN*	Asporin	Extra-cellular matrix.
*FN1*	Fibronectin	Extra-cellular matrix glycoprotein, binding to integrins.
*LAMA4, LAMB1, LAMC1*	Laminin	Major component of the basal lamina (basement membrane).
*AEBP1*	AE binding protein	Role in adipogenesis and smooth muscle differentiation.
*FLNA*	Filamin	Actin filaments link to membrane glycoproteins.
*MME*	Membrane Metalloendopeptidase	Type 2 transmembrane glycoprotein.
*FMOD*	Fibromodulin	Fibrillogenesis inhibition of collagen type 1 and 2.

## Data Availability

Not applicable.

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
