# Peer review of "Association between Expression of Connective Tissue Genes and Prostate Cancer Growth and Progression"

_ijms, 2023, doi:10.3390/ijms24087520_

Round 1

Reviewer 1 Report

This study investigates  connective tissue gene expression in relation to prognosis after prostate cancer therapy. 

General

While the study has good data, the poor presentation and language difficulties make it hard for the reader to follow. Many lengthy and ambiguous statements only make matters worse.  Commas have been used extensively where should not be.

Introduction:

This section could be improved with an overview on prostate cancer prevalence, morbidity and mortality both in the US and globally. Some statement on therapeutic approaches and prognosis would be helpful.

Material and methods :

It is unclear how sampling was done, the ages of patients, the types of samples etc.

Were there any controls for the expression of genes? 

Results:

The axes in the graphs are not clearly labeled.

NB: further comments are to be found in the tracked-in PDF document.

Author Response

Dear Reviewer, 

Thank you very much for your review and your comments that are very helpful, I have revised the paper according to your side notes. 

Introduction: I have modified introduction according to your comments 

Methodology: I have revised it according to your comments, as for the expression of genes, there was no control done unfortunately as we only had the raw data of the expression, we didn't do the calculation ourselves, I have put a sentence accordingly in the methodology. 

Results: Title has been change to describe the axes

Reviewer 2 Report

In the present study, the authors evaluated the association between genomic features of connective tissue and pejorative 14 clinical outcomes on radical prostatectomy specimens. Their results indicate that transcriptomic overexpression of connective tissue genes correlated to worse clinical features, such as ECE, clinically significant cancer and BCR, identifying the potential prognostic value of the gene signature of the connective tissue in prostate cancer. However, the data analysis in this paper is too simple, with no relatively innovative findings and lack of experimental data to support. Therefore, I recommend rejecting this paper unless additional experiments are conducted and additional data base, genomics and other analyses are added.

Q1: Why is there no concluding analysis in Part 2.2? The row spacing in Table 2 is different.

Q2: Why is there no relevant conclusion after analyzing ISUP Score and other indicators?

Q3: In figure 1B, the grouping legend overlaps the data.

Q4: In figure 2, the resolution of the picture is low. You are advised to upload a high-resolution picture. It is suggested to conduct 5-year PFS analysis for the clinical samples in Figure 1.

Q5: Better data processing software is recommended for data analysis, such as Prism, etc.

Q6: It is suggested to add relevant experimental data, such as Western blotting, qPCR, IHC and other experiments, to verify the different genes in Figure 1.

Author Response

Dear Reviewer, 

Thank you very much for your very important comments that are very helpful in this manuscript, I will try to answer your questions one by one: 

Q1: Thank you for your comment:

Regarding a conducting analysis, we were only able to access the raw data from RNA expression from Decipher company, and didn't have the manpower nor time to conduct additional experiments, the goal of the publication was more of a translational approach rather than a fundamental research paper. We have therefore changed the discussion and added your comment in our limitations. 

Q2: Thank you for your comment, sorry I am not sure if I understand this question, we showed results regarding overexpression of genes in different ISUP groups

Q3: Thank you for your comment, We have modified accordingly to your comment 

Q4: Thank you for your comment, We have redone an analysis and modified the graph according to your comments

Q5: Thank you very much for your comment, we completely agree on it and wished we had used one of the software at time of study, we are sorry we weren't able to use one of them at time of study 

Q6: Thank you very much for your comment, we are sorry we didn't have the human ressources to do additional experiments such as immunohistochemistry or Western Blotting, we only had clinicians on our publication and were shared the raw data from the Decipher company, we have added this in our limitations 

Reviewer 3 Report

The authors demonstrated well that prognosis of prostatic cancer and connective tissue genes. Their findings are sound very interesting.

Only minor changes are required before publication.

1. line 49, "adoptive" ?  or "adptive" 2. What is the definition of ISUP? Please describe. page 7, What is the Figure number? Please add.

update : 25 Jan. 2023

Major comments; 

1. This reviewer recommend to show the significance of overexpression of collagen 1 and collagen 3  genes by Sirius Red staining (to detect collagen deposition) or immunohistochemistry of collagen type 1 and type 3 using typical lymph node invasion samples and Extra-Capsular-Extension samples. These results provide strong support for the author's findings.

2. In addition, to show the direct evidence of the author's results, I recommend to examine whether overexpression of collagen1 and/or 3 and other connective tissue genes such as versican etc. promotes invasion and migration ability of prostatic cancer cell lines.

 Minor points,

1. What is LUM that is described in Abstract, Results section and Table 2 ? Is it Laminin? But, in Table 4, Laminin genes are described as LAMA4 or LAMB1 and also in abbreviation, described as LAM: Laminin.

Author Response

Thank you very much for your comments that have been very helpful in making this publication more impactful in our community; 

We will reply to your comments one by one: 

1: we changed adoptive to adaptive 

2: ISUP (International Society of UroPathologist): we gave a detailed description of it in the methodology 

3: I am sorry, Page 7, I am not sure of what you are talking about 

Major comments: 

Thank you so much for your interesting comment, we are sorry we couldn't make additional experiments as we only had clinicians in this study, but we added this in our limitations and perspectives as it could be an approach for another study 

Lum is Lumican, we added it in the abbreviations 

Round 2

Reviewer 1 Report

REVIEW REPORT IJMS-2187092Revised version by Patrick-Julien Treacy  et al.

Article title: Association between expression of connective tissue genes and prostate cancer growth and progression

Generally, the author as have adequately responded to the issues I raised earlier but there are still few discrepancies as indicated below:

Line 21; Node (LN) invasion and eBCR (early Biochemical Recurrence, defined as earlier than 3 years after write the abbreviation in brackets after the complete name

Line 182_’Regarding the PFS (Progression-Free Survival), out of the 111 patients with an expression’ -put the abbreviation in brackets after the full name.

Figure 2 legend is missing.

Line 254:  Moreover, in the TCGAp database, our chosen genes showed to be upregulated from 0,6;-change 0,6 to 0.6

Line 296: rephrase statement.

LINE 302: gy slides using the latest ISUP consensus. (International Society of Urolog ical Pathology Consensus (14)) to score the lesion:- -put the abbreviation in brackets after the full name.

Line 326: DAVID (Data- base for Annotation, Visualization and Integrated Discovery):- put the abbreviation in brackets after the full name.

Author Response

Thank you very much for your comments, 

We have done the modifications according to them!